# Structural basis for selectivity and antagonism in extracellular GPCR-nanobodies

Roman R. Schlimgen [1], Francis C. Peterson [1], Raimond Heukers [2], Martine J. Smit [2], John D. McCorvy [3] & Brian F. Volkman [1] ✉

G protein-coupled receptors (GPCRs) are pivotal therapeutic targets, but their complex structure poses challenges for effective drug design. Nanobodies, or single-domain antibodies, have emerged as a promising therapeutic strategy to target GPCRs, offering advantages over traditional small molecules and antibodies. However, an incomplete understanding of the structural features enabling GPCR-nanobody interactions has limited their development. In this study, we investigate VUN701, a nanobody antagonist targeting the atypical chemokine receptor 3 (ACKR3). We determine that an extended CDR3 loop is required for ACKR3 binding. Uncommon in most nanobodies, an extended CDR3 is prevalent in GPCR-targeting nanobodies. Combining experimental and computational approaches, we map an inhibitory ACKR3-VUN701 interface and define a distinct conformational mechanism for GPCR inactivation. Our results provide insights into class A GPCR-nanobody selectivity and suggest a strategy for the development of these new therapeutic tools.

Over 800 human G protein-coupled receptors (GPCRs) permit cells throughout the body to respond to a myriad of extracellular cues[1]. The fine-tuned control of these receptors has driven them to become the most abundant class of therapeutic targets[2]. However, only an eighth of human GPCRs have successfully been targeted with current therapeutic strategies[2]. While small molecules and peptides are most widely used to therapeutically target GPCRs, new molecules are needed to target the remaining receptors[3]. Monoclonal antibodies (mAbs), a successful class of biologic drugs, have been explored as a means to overcome challenges in the therapeutic targeting of GPCRs[2]. In the past 4 years, two GPCR-directed mAb drugs have been approved by the FDA (erenumab−CGRP-R, mogamulizumab−CCR4)[4–6]. These 150 kDa multi-chain proteins encode binding affinity and selectivity in a large, highly variable antigen binding site[7]. Despite some success, the expansive binding surface of a mAb is not always well suited for selective recognition of relatively small extracellular epitopes displayed by membrane-embedded GPCRs[8]. A recent evaluation of 407 anti-GPCR Abs showed that only 61% (248) were target-specific[9],

reinforcing the challenges associated with biologic drug development for GPCRs.

Variable domain heavy-chain-only immunoglobins (VHH, also known as nanobodies) have the potential to bridge the gap between mAbs and small molecules. These small (12–15 kDa), easily produced proteins use a smaller paratope of three complimentary-determining regions (CDRs) to bind GPCR epitopes that are more difficult to target with mAbs[10,11]. Approval in 2019 of the first nanobody drug, caplacizumab, a von Willebrand factor inhibitor, accelerated the development of this new class of therapeutic molecules[12]. Nanobodies targeting two GPCRs, the chemokine receptors CXCR4 and CX3CR1, are now in clinical trials[13,14]. To fully exploit the power of nanobodies as pharmacologic tools and potential drugs, it will be important to understand how nanobodies bind extracellular GPCR epitopes and alter receptor signaling. We will focus on the most abundant GPCR targets, class A GPCRs.

Of the >1900 solved nanobody structures, only 2% target class A GPCRs (Fig. 1a), and most of those bind the intracellular

¹Department of Biochemistry, Medical College of Wisconsin, Milwaukee, WI 53226, USA. ²Amsterdam Institute of Molecular and Life Sciences, Department of Chemistry and Pharmaceutical Sciences, Division of Medicinal Chemistry, Faculty of Science, Vrije Universiteit, 1081 HZ Amsterdam, The Netherlands. ³Department of Cell Biology, Neurobiology, and Anatomy, Medical College of Wisconsin, Milwaukee, WI 53226, USA. ✉e-mail: bvolkman@mcw.edu

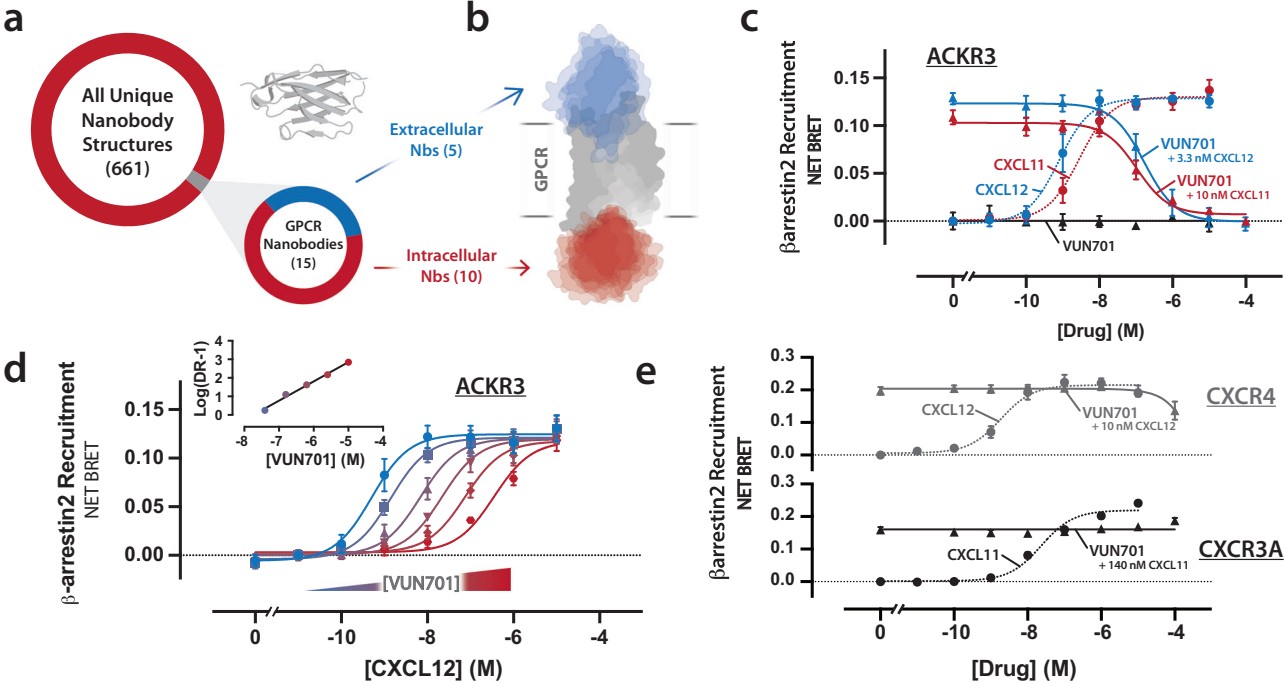

**Fig. 1 | Pharmacological characterization of the nanobody VUN701. a** Plot of all non-redundant nanobody structures in the Protein Data Bank with the gray region indicating the small proportion that targets GPCRs. The inlaid plot separates the non-redundant GPCR-targeting nanobodies based on their epitope (intracellular vs extracellular surface). **b** Schematic of the nanobody-accessible surfaces of a GPCR. All intracellular targeting nanobodies are shown in red as surface representations. The five extracellular targeting nanobodies with solved structures (JN241, Sb51, Nb29, Nb2, and NbE) are shown in blue as surface representations. **c** BRET-based β-arrestin2 recruitment assay of ACKR3 with endogenous ligands CXCL11 and CXCL12 (dashed lines). VUN701 competition assay with increasing VUN701 in the presence of indicated chemokine concentrations (solid lines). $N = 3$ biologically independent experiments plotted as mean +/- SEM. **d** Dose-dependent ACKR3 activation in the presence of increasing VUN701 with inlaid Schild-plot of the dosage response curves. $N = 3$ biologically independent experiments plotted as mean +/- SEM. **e** CXCR4 (gray) and CXCR3A (black) BRET-based β-arrestin2 assay with increasing CXCL12 and CXCL11 (dashed lines) or competition assay of chemokine with increasing WT VUN701 (solid lines). $N = 3$ biologically independent experiments plotted as mean +/- SEM. Source data are provided as a Source Data file.

surface of the receptor (Fig. 1b). Only two of these five anti-GPCR nanobodies that bind the extracellular surface are pharmacologically active[15–19]. These nanobodies, JN241 and NbE, are neutral antagonists for the apelin receptor and μ opioid receptor, respectively. JN241 has also been engineered to function as an apelin receptor agonist (JN241-Y)[15]. Several other nanobodies have been characterized as extracellular antagonists, but details of the nanobody-GPCR interface and the structural basis for receptor inhibition are lacking.

Using bioluminescence resonance energy transfer (BRET) assays for GPCR activation, we characterize the receptor specificity of a nanobody (VUN701) directed against the atypical chemokine receptor 3 (ACKR3). We establish VUN701 as a competitive ACKR3 inhibitor with extracellular therapeutic potential. Solving the solution structure of VUN701, we reveal an unusual motif that enables its inhibition. While uncommon in most nanobody structures, we find this distinctive motif is a frequent feature of GPCR targeting nanobodies. We map the inhibitory ACKR3-VUN701 interface and define a molecular mechanism by which ACKR3 and other GPCRs can be inhibited. These results provide structural insight into how GPCRs can be inhibited by nanobodies and suggest a strategy for the development of powerful tools with many uses.

## Results

### VUN701 selectivity and mechanism
Using a BRET assay to measure β-arrestin2 recruitment to ACKR3, we showed that CXCL11-dependent signaling was inhibited by VUN701 with an $IC_{50} = 105$ nM and $K_B$ of 23.3 nM ($pK_B = 8.64$; Fig. 1c and Supplementary Table 1). VUN701 also inhibited CXCL12 activation with an $IC_{50} = 157$ nM, corresponding to a VUN701 binding affinity ($K_B$) of

9.8 nM ($pK_B = 8.01$; Fig. 1c and Supplementary Table 2), consistent with previous results from CXCL12 radioligand displacement ($K_D = 10$ nM, $pK_D = 8.00$)[20].

Schild analysis[21] of CXCL12-induced β-arrestin2 recruitment to ACKR3 showed that VUN701 shifted CXCL12 potency in a concentration-dependent manner without altering the slope or maximum efficacy of CXCL12 β-arrestin2 recruitment (Fig. 1d, Supplementary Table 3). The Schild plot (Fig. 1d - inset) provided a VUN701 binding affinity to ACKR3 ($K_B$) of 18.2 nM, consistent with previous estimates ($pK_B = 7.74$).

CXCR4 or CXCR3, the closest paralogs of ACKR3, are receptors for CXCL12 and CXCL11, respectively. VUN701 failed to inhibit β-arrestin2 recruitment for either chemokine-GPCR pair at concentrations up to 10 μM (Fig. 1e, Supplementary Tables 1 and 2). Taken together, inhibition of CXCL11 and CXCL12 function in cell-based assays and Schild analysis with CXCL12 demonstrate that VUN701 functions as a selective, reversible competitive antagonist. The direct competition of VUN701 with CXCL11 and CXCL12 indicates that the ligands have overlapping, orthosteric epitopes.

### CDR3 of VUN701 is an extended hairpin
To understand how VUN701 is able to bind to the orthosteric site of ACKR3, we solved the solution structure of VUN701 by NMR (Fig. 2a and Supplementary Table 4, PDBID 8UEK). VUN701 adopts the conserved β-sandwich immunoglobulin fold observed in other nanobody structures with three variable loops that comprise the complimentary-determining regions (CDRs) and predominantly mediate binding to the target protein. An extended β-hairpin conformation observed for CDR3 was confirmed by the presence of strong cross-strand nuclear Overhauser effect (NOE) signals, low root mean square deviation

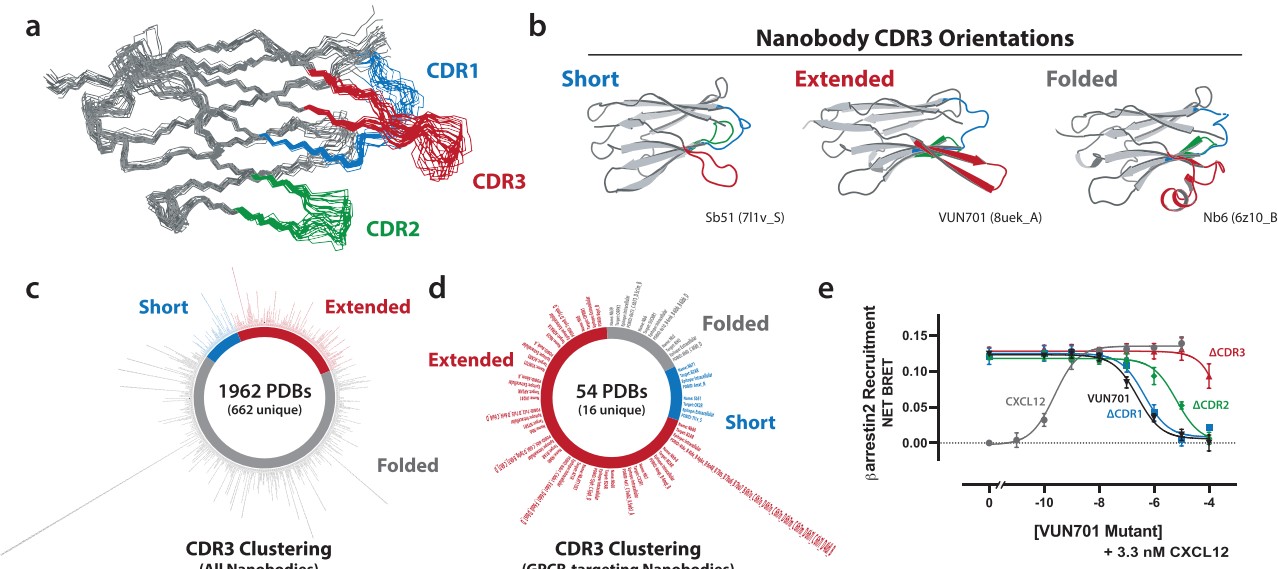

**Fig. 2 | Structural characterization of VUN701. a** NMR structural ensemble of VUN701 (PDB: 8UEK) with the three nanobody complimentary-determining regions (CDRs) colored. **b** Representative GPCR-targeting nanobodies belonging to the three CDR3 conformations: short (left), extended (middle), and folded (right) − complimentary-determining regions colored as in A. Nanobody names and PDB IDs are displayed below structures. **c** Plot of all 1962 nanobody structures in the PDB, colored based on the distribution of defined CDR3 conformations. Of the 1962 structures, there are 662 non-redundant structures. PDB IDs of each nanobody structure are placed along the outside of the pinwheel, sorted by their CDR3 type. Redundant nanobody structures are placed on the same line. **d** Plot of all 54 GPCR-targeting nanobody structures in the PDB, colored based on the distribution of CDR3 conformations. 16 represent non-redundant structures. PDB IDs of each nanobody structure are placed along the outside of the pinwheel, sorted by their CDR3 type with their name, GPCR target, and surface epitope. Redundant nanobody structures are placed on the same line. **e** ACKR3 BRET-based β-arrestin2 assay with increasing CXCL12 (gray) or competition assay of 3.3 nM CXCL12 with increasing WT VUN701 (black), and CDR variants ΔCDR1 (blue), ΔCDR2 (green), and ΔCDR3 (red) illustrating the alteration in $IC_{50}$ due to deletion of CDR important contacts. $N = 3$ biologically independent experiments plotted as mean +/- SEM. Source data are provided as a Source Data file.

(RMSD) values, and high heteronuclear NOE ratios for residues 91−101 and 105−109 (Supplementary Fig. 1).

VUN701's extended CDR3 is an unusual feature in the landscape of solved nanobody structures. In an all-by-all structural comparison, the 1962 solved nanobody structures fell into three CDR3 classes: folded-back (66%), extended (25%), or short (9%) (Fig. 2b, c, Supplementary Data 1). Interestingly, when considering only class A GPCR-targeting nanobodies, a different CDR3 distribution was observed, with almost 70% in the extended class (Fig. 2d, Supplementary Data 2). We speculated that the extended CDR3 "β-finger" of VUN701 participates directly in high-affinity ACKR3 binding and replaced each CDR of VUN701 to assess their respective contributions (Supplementary Fig. 2). While the ΔCDR1 and ΔCDR2 mutants retained the ability to bind and inhibit ACKR3, the ΔCDR3 mutant resulted in a near-total loss of inhibition (Fig. 2e). The β-finger's functional importance and enrichment of extended CDR3s in nanobodies that bind GPCRs prompted the hypothesis that VUN701 binds the orthosteric pocket of ACKR3, interacting in a manner that mimics the N-termini interactions of the chemokine ligands[22].

**Mapping the VUN701-ACKR3 interface**

Using AlphaFold2 and molecular dynamics simulations, we modeled ACKR3 in complex with its natural ligand CXCL12 (Supplementary Fig. 3a). The AF2/MD model generated using this approach reproduced the ACKR3-CXCL12-Fab-Nanobody structure solved by cryo-EM[23] with a backbone RMSD of 2 Å (Supplementary Fig. 3b). A model of the ACKR3-VUN701 complex prepared by the same method (Fig. 3a) formed an extensive protein-protein interface that includes residues from all three CDRs and occupies the ACKR3 orthosteric site. None of the VUN701 residues altered in the ΔCDR1 construct (T28$^{CDR1}$, F29$^{CDR1}$, S30$^{CDR1}$, L31$^{CDR1}$; Supplementary Fig. 2) were in contact with ACKR3, but H32$^{CDR1}$ was positioned to form a cation-π interaction with F285$^{ECL3}$ of ACKR3. In contrast, D56$^{CDR2}$, the ΔCDR2 substitution, made no stable

contacts with ACKR3 that would explain its functional importance. The extended CDR3 β-finger buried itself into ACKR3's orthosteric pocket, where K100 was positioned between ACKR3's D275$^{6x58}$ and E290$^{7x27}$, and R103$^{CDR3}$ was poised to form a salt bridge with D179$^{4x61}$.

To assess the importance of the predicted VUN701 interactions, each CDR residue was replaced with alanine. None of the substitutions significantly destabilized the structure based on nanoDSF measurements of thermal unfolding, and the $T_m$ value for most variants was within 3 °C of VUN701 (71.0 °C) (Supplementary Table 5). Using the BRET assay for CXCL12-stimulated β-arrestin2 recruitment (Fig. 1c), we compared $IC_{50}$ values for each VUN701 variant to assess the change in inhibitory potency. Several alanine substitutions were found to have altered affinites compared to Wild-type (WT) VUN701 (Fig. 3b, Full data in Supplementary Fig. 4). To easily visualize the impact of alanine substitutions on VUN701's functional affinity, the change in $\log(K_B)$ vs WT was calculated (Fig. 3c; Supplementary Table 6). Five substitutions caused a significant loss of VUN701 binding: H32$^{CDR1}$, D56$^{CDR2}$, K100$^{CDR3}$, R103$^{CDR3}$, and F106$^{CDR3}$. Among them, the K100A$^{CDR3}$ mutation was the most deleterious, resulting in an affinity reduction below the detection limit of the assay. Several CDR3 substitutions (G102A$^{CDR3}$, D104A$^{CDR3}$, R107A$^{CDR3}$) increased VUN701 potency while also reducing solubility in phosphate-buffered saline (PBS) buffer by a factor of ~6.

To assess the importance of ACKR3 contacts observed in the model, we selected 14 charged and aromatic residues that line the chemokine binding pocket for alanine substitution (Fig. 3e). Except for three variants with low expression in HEK293T cells (Supplementary Fig. 5), we determined the functional effect of ACKR3 substitutions as reflected in the change of $EC_{50}$ for CXCL12-induced β-arrestin2 recruitment (Fig. 3d). Five of the substitutions had little or no effect on CXCL12 potency while the other six shifted the $EC_{50}$ by 2- to 100-fold. Using the 90% effective concentration of CXCL12 determined for each of the 11 ACKR3 variants, inhibitory potencies of VUN701 were measured (Supplementary Fig. 6). $K_B$ values were computed for each

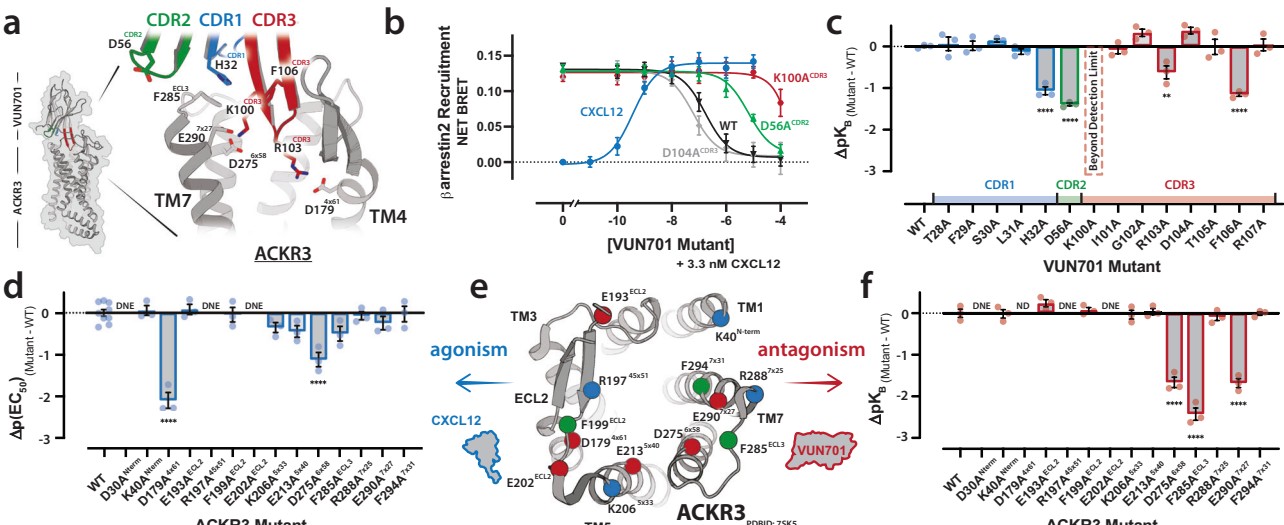

**Fig. 3 | ACKR3-VUN701 β-arrestin2 dependent paratope mapping. a** Model of the ACKR3-VUN701 complex with zoomed view of the orthosteric interactions between VUN701 and ACKR3. Intermolecular interactions are shown as sticks and are colored according to their CDR location. **b** ACKR3 BRET-based β-arrestin2 assay with increasing CXCL12 (blue) or competition assay of 3.3 nM CXCL12 with increasing WT VUN701 (black), and VUN701 variants D56A (green), K100A (red), and D104A (gray) illustrating the alteration in IC$_{50}$ due to CDR mutations. $N = 3$ biologically independent experiments plotted as mean +/- SEM. **c)** Plot of the change in pK$_B$ values for VUN701 CDR alanine mutants ($N = 3$ biologically independent experiments plotted as mean +/- SEM. Residues with significant alterations in binding marked with asterisks (Ordinary one-way ANOVA). K100A mutation shifted VUN701 binding beyond the limit of detection. **d** Plot of change in pEC$_{50}$ values for CXCL12 on orthosteric alanine ACKR3 mutants ($N = 3$ biologically independent experiments plotted as mean +/- SEM, Ordinary one-way ANOVA). **e)** Cartoon representation of ACKR3's orthosteric pocket with residues probed in alanine-scanning marked with circles based on charge (red – negative; blue – positive) or aromaticity (green). **f** Plot of changes in pK$_B$ values for VUN701 on orthosteric alanine ACKR3 mutants ($N = 3$ biologically independent experiments plotted as mean +/- SEM, Ordinary one-way ANOVA). $^{ND}$Mutant shifted CXCL12's EC$_{50}$ beyond accurate determination of VUN701's IC$_{50}$; $^{DNE}$Mutant did not express. $P < 0.0001$ (****); $P < 0.031$. All receptor residues are numbered in GPCRdb numbering format. Source data are provided as a Source Data file.

mutant to account for the change in CXCL12 potency and reflect the effect of each substitution on VUN701 binding (Fig. 3f; Supplementary Table 2). In contrast to CXCL12, only three of the alanine substitutions diminished VUN701 binding: D275$^{6x58}$, F285$^{ECL3}$, E290$^{7x27}$. A positive correlation between the number of intermolecular contacts made at each residue versus the shift in K$_B$ resulting from alanine substitution (Supplementary Fig. 7) suggests that the computational model of this complex accurately predicts important elements of VUN701-ACKR3 recognition.

## Mechanism of ACKR3 antagonism

Structural comparisons of ACKR3 bound to the chemokine agonist CXCL12 and the nanobody antagonist VUN701 could reveal key contacts or conformational changes for inactivation of the receptor. The largest difference between the VUN701-ACKR3 model and the CXCL12-ACKR3 cryoEM structure (PDB: 7SK5) was a ~5 Å shift of transmembrane helix 7 (TM7) (Fig. 4a). Binding of VUN701 appears to position CDR1 and CDR2 directly above TM7, forming a lid that would prevent TM7 from occupying the location observed in the CXCL12-ACKR3 structure. The CXCL12-induced upward shift of TM7 engages a conserved network of polar side chains that stabilizes the active conformation. This network includes the NPxxY motif in TM7 which interacts with D90$^{2x50}$ (Fig. 4b). VUN701 binding disfavors the upward shift of TM7 that is required for engagement of the conserved NPxxY activation motif (Supplementary Fig. 8). To assess the validity of the TM7 shift suggested by the VUN701-ACKR3 model, we scanned the TM6-TM7 interface for side chain contacts that were significantly altered relative to the CXCL12-ACKR3 cryoEM structure as potential sites for the introduction of a disulfide bond that would lock ACKR3 into the inactive conformation. The disulfide-compatible Cβ-Cβ distance of 4.8 Å between A271$^{6x54}$ and L293$^{7x30}$ in the VUN701-ACKR3 model stretches to 7.4 Å in the CXCL12-ACKR3 cryoEM structure (Fig. 4c). We introduced cysteine at each position and compared the

CXCL12-induced activation of WT and variant ACKR3 receptors (Fig. 4d). ACKR3 activation was substantially reduced upon the introduction of cysteine at both positions, while neither single substitution had a significant impact on receptor function.

The addition of the A271C$^{6x54}$ L293C$^{7x30}$ disulfide resulted in a 50% increase in expression, as measured by Rluc counts, as well as a small increase in the basal β-arrestin2 recruitment to ACKR3 (Supplementary Fig. 9). For all cysteine mutants, the CXCL12 EC$_{50}$ was largely unaffected suggesting that the chemokine binding interface remains intact (Fig. 4d, Supplementary Table 2). Unlike CXCL12, VUN701's potency was significantly decreased and it acted as an inverse agonist for ACKR3 β-arrestin2 recruitment (Supplementary Fig. 9). These results suggest that an engineered TM6-TM7 disulfide restricts the conformational changes necessary for full ligand-induced activation.

## VUN701 selectivity

VUN701 inhibits ACKR3 with nanomolar potency but has no effect on CXCR3 or CXCR4 activation at concentrations up to 100 µM (Fig. 1e). Analysis of sequence conservation across all chemokine receptors shows that the transmembrane helices and residues lining the orthosteric pocket are more conserved than the N- and C-termini, ECL2, or ECL3 (Fig. 5a). Examining the currently solved structures of chemokine receptors bound to small molecule antagonists, we observed that these orthosteric antagonists exclusively target conserved receptor regions. Each small molecule displayed a remarkably similar interaction fingerprint, engaging in an average of 77 contacts with a common set of orthosteric residues in TM1, TM2, TM3, TM6, and TM7 (Fig. 5b). In contrast, VUN701 formed 175 contacts with a much larger interface in our model that includes the orthosteric pocket and ACKR3 residues in the low-conservation N-terminus, ECL2, and ECL3 (Fig. 5c). Notably, key CDR3-interacting residues D275$^{6x58}$ and E290$^{7x27}$ (Fig. 3a) are also present in CXCR3 and CXCR4 (Fig. 5d), receptors that share the chemokine ligands CXCL11 and CXCL12, respectively, with

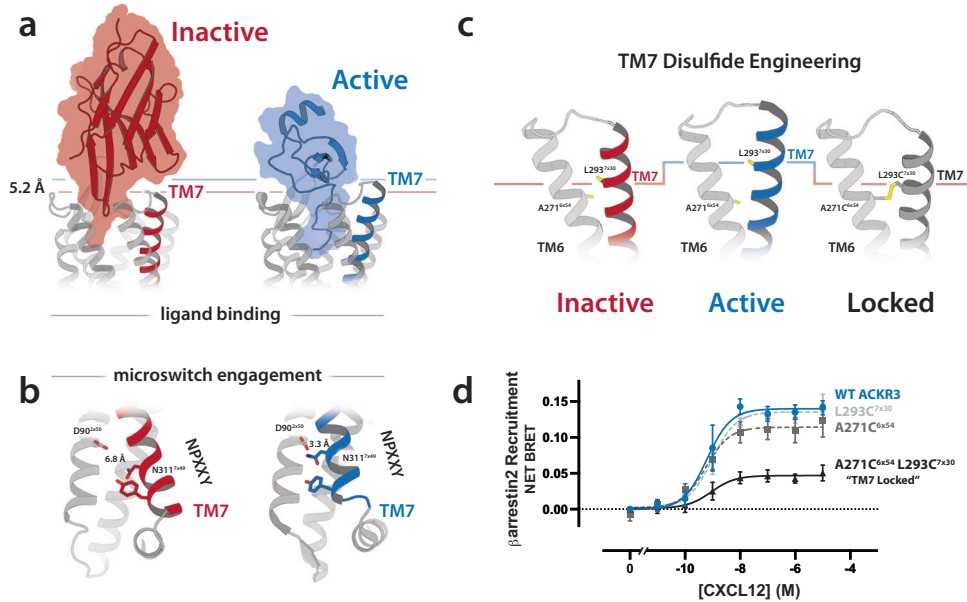

**Fig. 4 | Structural mechanism of VUN701's antagonism. a** Comparison of VUN701-bound ACKR3 model (red) and CXCL12-bound ACKR3 structure (blue) (PDB: 7SK5) indicating the 5.2 Å upward shift of TM7 promoted by CXCL12 binding. VUN701 CDR1 and CDR2 loops disfavor the TM7 shift. **b** Intracellular view of the alignment, highlighting the stabilization of the NPxxY motif (D90$^{2x50}$, N311$^{7x49}$, Y315$^{7x53}$) in an inactive conformation upon VUN701 binding. **c** Extracellular view of the structural alignment to visualize the top of TM6/TM7, with AF2 model showing the location of engineered disulfides to prevent TM7 upward shift. Residues changed to cysteine (A271$^{6x54}$, L293$^{7x30}$) are shown in yellow as alpha and beta carbons. **d** WT ACKR3 BRET-based β-arrestin2 assay with increasing CXCL12 (blue) or ACKR3 with engineered disulfide to immobilize the upward TM7 transition (black). Single disulfide mutants are shown in gray as dashed lines. $N = 3$ biologically independent experiments plotted as mean +/- SEM. Source data are provided as a Source Data file.

ACKR3. Thus, while the contacts that VUN701 makes by docking the CDR3 β-hairpin in the ACKR3 orthosteric pocket are important for binding affinity (Fig. 3f), they cannot alone encode its selectivity for ACKR3 relative to CXCR3 or CXCR4.

The ACKR3 substitution that had the greatest effect on VUN701 binding, F285$^{ECL3}$, is highly variable in the chemokine receptor family and corresponds to R288$^{ECL3}$ and Q272$^{ECL3}$ in CXCR3 and CXCR4, respectively (Fig. 5d). For this reason, we speculated that this residue may encode a high degree of selectivity. However, mutation of CXCR3's R288$^{ECL3}$ or CXCR4's Q272$^{ECL3}$ to mimic ACKR3's F285$^{ECL3}$ did not enable VUN701 binding at concentrations up to 100 μM (Supplementary Fig. 10, Supplementary Table 7 and 8). Chimeric constructs of CXCR3 and CXCR4 containing ACKR3's entire ECL3 also exhibited no VUN701 binding (Supplementary Fig. 10, Supplementary Table 7 and 8).

Superimposition of the VUN701-ACKR3 model with CXCR3 and CXCR4 suggested that, in addition to ECL3, residues of ECL2 and TM5 also may be incompatible with VUN701 binding, particularly due to clashes with CDR1 (Supplementary Fig. 11). To test the hypothesis that CDR1 contributes to VUN701 selectivity, we measured inhibition of β-arrestin recruitment by the ΔCDR1 variant (Supplementary Fig. 2), which binds ACKR3 with a $K_B$ of 32 nM ($pK_B$ = 8.49; Fig. 2e). While the ΔCDR1 variant exhibited no inhibitory activity toward CXCR3, this substitution now imparted CXCR4 inhibition to VUN701 with low micromolar potency ($IC_{50}$ = 3.94 μM; $K_B$ = 948 nM; $pK_B$ = 6.02; Fig. 5e, f, Supplementary Table 10). Taken together, these results demonstrate that the extended CDR3 β-hairpin of VUN701 is responsible for high-affinity interactions in the conserved orthosteric pocket whereas receptor selectivity is largely encoded by distal contacts between VUN701 and ECL residues unique to ACKR3.

## Discussion

Their compact size and intricate binding epitopes position nanobodies to fill the therapeutic niche in GPCR drug development where small

molecules and antibodies fall short. Despite the explosion of the nanobody field relatively few of the >15,000 unique nanobodies generated to date target GPCRs[24]. To expand the utility of this important biologic drug platform in the GPCR field we defined the structure-function relationships for VUN701 and its target, the atypical chemokine receptor ACKR3.

In the VUN701 NMR structure, CDR3 adopts a stable β-hairpin that extends from the conserved Ig domain fold (Fig. 2b). While extended CDR3s are highly enriched among GPCR-targeting nanobodies (Fig. 2c, d), the majority of the reported structures involve a nanobody ligand that binds an intracellular surface of the receptor. The JN241-apelin receptor complex and the NbE-mu opioid receptor complex are currently the only examples of a nanobody antagonist – both with an extended CDR3 bound in the extracellular pocket[15,19]. From the available data, it appears that all pharmacologically active GPCR nanobodies possess an extended CDR3. Whereas traditional antibody recognition involves enveloping the antigen in a cleft surrounded by the hypervariable loops or binding a large, flat antigen surface, globular monovalent molecules like nanobodies lack this cleft[25]. Instead, anti-GPCR nanobodies use the extended CDR3 to mimic GPCR ligand or transducer binding by inserting into an extracellular or intracellular pocket. This has implications for the design of anti-GPCR nanobodies and the construction of synthetic libraries for screening, where the length and composition of CDR3 should be the primary focus.

Compared to small-molecule GPCR ligands, nanobodies like VUN701 can engage a larger number of intermolecular contacts. Beyond the ligand binding pocket, VUN701's CDR1 and CDR2 interact with ACKR3's N-terminus, ECL2, and ECL3 regions, encoding the details of molecular recognition across a substantial protein-protein interface. Such an expansive range of contacts enables a relatively high level of specificity. Emphasizing this, we found that CXCR3 and CXCR4 could not be antagonized by VUN701 despite the presence of key ACKR3-VUN701 contacts (D275$^{6x58}$, E290$^{7x27}$) in the orthosteric binding pockets of all three receptors. Furthermore, chimeric CXCR3 and CXCR4 receptors containing the ACKR3 ECL3 failed to bind VUN701

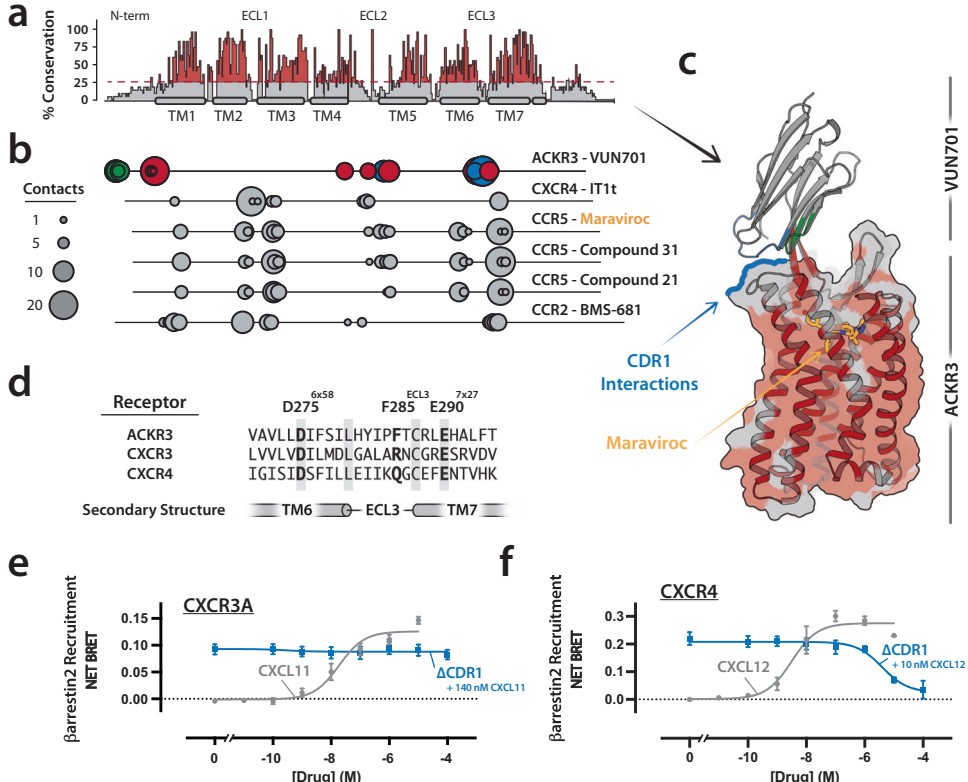

**Fig. 5 | Selectivity determinants of VUN701. a** Amino acid sequence conservation of all chemokine receptor sequences with the location of the N-terminus, TM helices, and extracellular loops indicated. Sequences were aligned based on GPCRDB nomenclature and conservation was scored based on the level of complete sequence conservation at each position using UGENE[55]. Positions with greater than one standard deviation were plotted in red. **b** Linear array of intermolecular contacts aligned along sequence conservation plot indicating the contacts between previously described small molecule antagonists targeting chemokine receptors and VUN701's intermolecular contacts. VUN701 contacts are colored according to the CDR enabling contacts (CDR1−blue, CDR2−green, CDR3−red). **c** Model of VUN701-ACKR3 complex with superimposition of the CCR5 antagonist Maraviroc (orange). Amino acid positions with greater than one standard deviation in

sequence conservation are plotted in red. Relative to Maraviroc and other small molecule chemokine receptor inhibitors, VUN701 engages a larger surface area that includes regions of low sequence conservation. **d** Sequence alignment of the ACKR3 TM6, ECL3, and TM7 regions with the same regions of CXCR3 and CXCR4. Residues found to be important in facilitating VUN701 interactions are bolded. Conserved residues are shown with a gray background. ACKR3's D275[6x58] and E290[7x27] are fully conserved. F285[ECL3] has no sequence conservation. CXCR3 (**e**) and CXCR4 (**f**) BRET-based β-arrestin2 recruitment assay with increasing chemokine (gray) or competition assay of indicated chemokine with increasing VUN701 ΔCDR1 (blue). $N = 3$ biologically independent experiments plotted as mean +/- SEM. Source data are provided as a Source Data file.

(Supplementary Fig. 10). However, the ΔCDR1 substitution conferred CXCR4 antagonist function on VUN701 while preserving its ability to inhibit ACKR3 (Figs. 5f and 2e), illustrating the importance of CDR1 and CDR2 contacts for encoding ACKR3 specificity. These results demonstrate the feasibility of generating multi-specific anti-GPCR nanobodies based on structural information and modeling, with a wide range of potential applications in medicine and biotechnology.

Like many anti-GPCR nanobodies that bind an extracellular epitope, VUN701 is an antagonist of its target, ACKR3[11]. Using BRET and molecular modeling we showed that specific CDR3 contacts with the orthosteric pocket are required for its antagonist activity. Using charged and hydrogen-bonding interactions, VUN701 binding sterically blocks an upward shift of TM7 that is required for engagement of the conserved NPxxY activation motif. Conservation of this intracellular NPxxY motif (D90[2x50], N311[7x49], and Y315[7x53]) across the majority (>70%) of class A GPCRs suggests that VUN701's NPxxY disruption mechanism could be employed with other receptors[26]. This hypothesis was supported by the demonstration that VUN701 ΔCDR1 also inhibits β-arrestin2 recruitment for the classical GPCR, CXCR4 (Supplementary Fig. 5f). Moreover, contacts between the extended CDR3 β-hairpin and the ACKR3 ligand binding pocket could be altered to expand the VUN701 functional landscape. Structure-guided engineering of the CDR3 motif may enable the rapid development of specific agonists,

partial agonists, and inverse agonist nanobodies against ACKR3 or other GPCRs. As a proof of concept for this approach, a nanobody targeting the apelin receptor (APLNR) was converted into an agonist with an amino acid insertion into the CDR3[15]. Several other anti-GPCR antagonist nanobodies targeting the orthosteric site have been described[11]. We speculate that most of these also employ an extended CDR3, similar to VUN701, to enable high affinity GPCR interactions. For all of these molecules, the extended CDR3 can be modified to introduce new functionalities and can potentially act as an activation or inactivation switch.

## Methods
### Expression and purification of VUN701
The sequence of VUN701 was codon-optimized for *E. coli* expression and ordered from GenScript. VUN701 was cloned into a pET28a-6xHis-SUMO3 vector and expressed in BL21 DE3 *E.coli*. Amino acid substitutions were made using Quikchange mutagenesis (Product Number: 200524 – Agilent, primer sequences provided in Supplementary Data 3). Cells were expressed at 37 °C in Luria-Bertani (LB) medium and induced with 1 mM isopropyl-β-D-thiogalactopyranoside (IPTG) at an OD600 of 0.6. Cultures continued to grow for 5 and a half hours before bacteria were pelleted by centrifugation and stored at −20 °C. For uniform labeling with [15]N and [13]C, cells were grown in M9 minimal

media containing $^{15}$N-ammonium chloride as the sole nitrogen source and $^{13}$C-glucose as the sole carbon source. Bacterial pellets were resuspended in ~20 mL of Buffer A (50 mM Na$_2$PO$_4$ (pH 8.0), 300 mM NaCl, 10 mM imidazole, 1 mM phenylmethylsulphonyl fluoride (PMSF), and 0.1% (v/v) 2-mercaptoethanol (BME)) per pellet and lysed via sonication. Lysed cells were clarified at 18,000 x g and the supernatant was discarded. Pellets were resuspended by sonication in ~ 20 mL of Buffer AD (6 M guanidinium, 50 mM Na2PO4 (pH 8.0), 300 mM NaCl, 10 mM imidazole) and spun down at 18,000 x g for 20 min. Using an AKTA-Start system (GE Healthcare), supernatant was loaded onto a Ni-NTA column equilibrated in Buffer AD. The column was washed with Buffer AD, and proteins were eluted using Buffer BD (6 M guanidinium, 50 mM sodium acetate (pH 4.5), 300 mM NaCl, and 10 mM imidazole). Proteins were refolded overnight via drop-wise dilution into a 10-fold greater volume of Refold Buffer (50 mM Tris (pH 7.6), 150 mM NaCl) with the addition of 35 mM cysteine, and 0.5 mM cystine. Refolded protein was concentrated in an Amicon Stirred Cell concentrator (Millipore Sigma) using a 10 kDa membrane. Concentrated protein was added to 6-8 kDa dialysis tubing with the addition of ULP1 to cleave the N-terminal 6xHis-SUMO3-tag and dialyzed at 25 °C against Refold Buffer overnight. The AKTA-Start system was used to load the cleaved protein onto a Ni-NTA column equilibrated in VUN701 Buffer A (Refold Buffer + 10 mM Imidazole). The column was washed with VUN701 Buffer A, and the protein was eluted using VUN701 Buffer B (Refold Buffer + 500 mM Imidazole). VUN701 underwent four rounds of dialysis in 5 mM ammonium bicarbonate, lyophilized, and stored at −80 °C for further use. Purity and identity of VUN701 was confirmed by electrospray ionization mass spectrometry using a Thermo LTQ instrument and SDS-PAGE with Coomassie staining.

## VUN701 Nuclear Magnetic Resonance (NMR)
NMR experiments were performed on a Bruker DRX 600 equipped with a $^1$H/$^{15}$N/$^{13}$C TCI cryoprobe and a SampleJet robot for automated NMR screening of compounds in 96-well sample racks. NMR samples contained 775 μM $^{15}$N/$^{13}$C VUN701 with the addition of 5% D$_2$O, 0.02% NaN$_3$, and 25 mM deuterated 2-ethanesulfonic acid (MES) at pH 6.8. $^1$H, $^{15}$N, and $^{13}$C resonance assignments for VUN701 were obtained at 298 K using the following experiments acquired with TopSpin 3.6.7: $^{15}$N-$^1$H HSQC[27] 3D HNCA[28,29] 3D HNCO[28,30], 3D HN(CO)CA[28], 3D HN(CO)CACB, 3D HNCACB, 3D HN(CA)CO, 3D HCCONH, 3D SE C(CO)NH[31], and 3D HCCH TOCSY[32]. Distance restraints used in structure calculation were derived from $^{15}$N-edited and $^{13}$C-edited NOESY spectra collected with 80 msec delay times. NMR data was processed with NMRPipe 9.6[33] and XEASY 1.3.13[34] was used for resonance assignments and analysis of spectra. Backbone ϕ and ψ dihedral angle constraints and chemical shift guided predictions of secondary structure were generated from shifts of the $^1$H, $^{13}$C$_\alpha$, $^{13}$C$_\beta$, $^{13}$C', and $^{15}$N nuclei using the program TALOS + [35]. Distance and dihedral restraints were used to generate initial NOE assignments and preliminary structures with the NOEAS-SIGN module of CYANA 3.0[36]. Complete structure determination was undertaken as an iterative process of correcting and assigning NOEs and running structure calculations with CYANA[36]. The 20 CYANA conformers with the lowest target function were further refined by a molecular dynamics protocol in explicit solvent with XPLOR-NIH[37,38]. Heteronuclear NOE values were measured from an interleaved pair of two-dimensional $^{15}$N-$^1$H sensitivity-enhanced correlation spectra recorded with and without a 5-s proton saturation period.

## VUN701 mutant solubility measurements
Lyophilized VUN701 and VUN701 mutants were resuspended in PBS to a concentration of 20 mg/mL. Samples were centrifuged at 13,000 × g for 5 min to clarify. The clarified protein was quantified using the measured absorbance at 280 nm and a calculated molar extinction coefficient based on the amino acid sequence[39]. WT VUN701 was found to be soluble at concentrations >60 mg/mL. I101A, G102A, D104A,

T105A, and R107A mutations in CDR3 decreased solubility in PBS to a maximum of 10 mg/mL.

## VUN701 mutant thermal denaturation
50 μM of purified VUN701 and variants in 50 mM MES, pH 6.8 were analyzed using nanoDSF on the Nanotemper Prometheus NT.48. Protein samples and a buffer control were placed in triplicate into single use standard capillaries and heated from 20–95 °C with a heating rate of 0.5 °C/min. Samples were excited at 280 nm and the F350:F330 ratio was monitored during the run to determine an apparent melting temperature (T$_M$) for each sample.

## BRET experiments
BRET experiments were performed to provide a direct and high-throughput readout of receptor activity in response to functional ligand mutants. The full-length sequences of ACKR3 and CXCR4 were cloned into a pcDNA3.1 vector to include an N-terminal HA-FLAG tag and a C-terminal IDTG linker preceding the Rluc8 gene. Amino acid substitutions were made using Quikchange mutagenesis (Product Number: 200524 – Agilent, primer sequences provided in Supplementary Data 3). A pcDNA3.1 vector containing β-arrestin2 with an N-terminal Venus tag was also used in this assay. HEK293T cells (ATCC - CRL-3216) maintained in Dulbecco's modified Eagle's medium (DMEM) supplemented with 10% fetal bovine serum (FBS) were transiently transfected with 0.3 μg of receptor-Rluc8 and 5 μg of Venus β-arrestin at ~50% confluency using TransIT-293 (Mirus). At 24 h post-transfection, cells were resuspended in PBS supplemented with 0.1% glucose and plated in 96 wells plates at a density of 100,000 cells/well in a total of 60 μL. After an additional 1 h at 37 °C, cells were stimulated with Coelenterazine-H to a final concentration of 5 μM and incubated for 5 min. Ligands suspended in PBS supplemented with 0.1% glucose were added at concentrations from 10 pM to 100 μM. WT-ACKR3:CXCL12 was included in each plate as a positive control and BRET (540/480 nm) signals were measured after 10 min on the Mithras LB940 (Berthold Technologies) using MicroWIN2010 5.19. Measurements on individual plates were performed in duplicate for each ligand concentration. Data were analyzed with a fixed slope (Hill Slope = 1), nonlinear fit to create a dose-response curve in GraphPad Prism 9 or 10 (Graphpad Software Inc., San Diego, CA). Data from three independent plates were used in the analysis and calculation of standard error. All data were baseline corrected to wells collected in the absence of functional ligands. VUN701 binding constants (K$_B$) using a single concentration of the agonist chemokine were calculated using the Cheng-Prusoff equation[40,41]. The VUN701 binding constant (K$_B$) determined from a Schild analysis with CXCL12 was measured in GraphPad Prism 9[21].

(1)   Cheng-Prusoff Equation:

$$K_B = \frac{IC_{50}}{1 + (\frac{[Agonist]}{EC_{50}})}$$

CXCL12 used in this assay was purchased from Protein Foundry, LLC.

## Bioinformatic structural nanobody comparison
Complete amino acid sequences and corresponding PDB identities were extracted from the integrated nanobody database for immunoinformatics (INDI) for all nanobodies with solved structures (1962)[24]. Using a python script, all PDB files were extracted from the Protein Database and secondary structure was calculated using *define secondary structure of proteins* (DSSP) algorithm. DSSP was used to validate each protein fold and to find trends in nanobody CDR structure. After manual removal of all files with missing structural elements, incorrect folds, or redundant data, 662 structure files were aligned to view the distribution of complimentary determining regions (CDRs).

The RMSD of each nanobody was calculated against all other nanobodies, creating a 662 × 662 matrix to visualize nanobody structural trends. Three major nanobody populations emerged. Manual curation fixed any outliers in the clustering resulting from differences in nanobody sequence, purification tags, or CDR length. Visual analysis of the clusters indicated large variability in nanobody CDR3 positions, supported by the DSSP secondary structure analysis.

### ACKR3 molecular modeling

Molecular models of the ACKR3-CXCL12 and ACKR3-VUN701 complexes were generated with AlphaFold 2 using the ColabFold platform on Google Colaboratory[42,43]. The sequences of human ACKR3 and human CXCL12α were obtained from UniProt (P25106 and P48061)[44].

All sequences are given below:

>hACKR3
MDLHLFDYSEPGNFSDISWPCNSSDCIVVDTVMCPNMPNKSVLLYTLSFIYIFIFVIGMIA
NSVVVWVNIQAKTTGYDTHCYILNLAIADLWVVLTIPVWVVSLVQHNQW
PMGELTCKVTHLIFSINLFGSIFFLTCMSVDRYLSITYFTNTPSSRKKMVRR
VVCILVWLLAFCVSLPDTYYLKTVTSASNNETYCRSFYPEHSIKEWLIGMEL
VSVVLGFAVPFSIIAVFYFLLARAISASSDQEKHSSRKIIFSYVVVFLVCWLP
YHVAVLLDIFSILHYIPFTCRLEHALFTALHVTQCLSLVHCCVNPVLYSFIN
RNYRYELMKAFIFKYSAKTGLTKLIDASRVSETEYSALEQSTK

>hCXCL12
KPVSLSYRCPCRFFESHVARANVKHLKILNTPNCALQIVARLKNNN
RQVCIDPKLKWIQEYLEKALNK

>VUN701
QVQLVESGGGLVQAGGSLRLSCAASGSTFSLHLMGWYRQAPGKQ-
REVVATSGSGGDTNYADSVKGRFTISRDNDKNTVDLQMNNLKPEDTA-
DYYCRAQQKIGRDTFRDYWGQGTQVTVSS.

Sequences were input into ColabFold in a hetero-oligomer format and the program was run without structural templates to enhance output heterogeneity. The sequence alignment used MMseqs2 and UniRef90, and the value of ipTM was used to rank the five outputted models against one another. The top-ranked model was manually inspected before placement into a synthetic biological membrane containing Cholesterol, POPA, DDPC, DOPC, POPE, and POPS in a 6:2:2:8:6:1 ratio using Charmm-GUI[45–50]. Water, along with sodium and chloride ions at a final concentration of 150 mM was then added to the system (Supplementary Table 10). After assembly, the system was downloaded in GROMACS format. The resulting model underwent 6 rounds of equilibration at 310 K before a 1 microsecond all-atom molecular dynamics (MD) simulation using GROMACS 2021.2[51,52] according to 45. The above protocol was run three times with each complex, and the highest scoring models were used in these studies. The models displayed herein are all taken at 500 ns of the simulation, modeled in PyMOL 2.4.2. All receptor residues numbered in GPCRdb numbering format[53]. Input and output MD models are provided in Supplementary Data 4 and 5, respectively. MD reproducibility checklist provided in Supplementary Data 6.

### Antagonist contact mapping

Structures for CKRs bound to small molecule antagonists were gathered from literature review. Structures included the CXCR4 antagonist IT1t (3OE9), the CCR2 antagonist BMS-681 (5T1A), and the CCR5 antagonists Maraviroc (4MBS), Compound 31 (6AKY), and Compound 21 (6AKX). All PDB files were downloaded from the RCSB PDB. Intermolecular (residue-ligand) contacts for each prepared ligand-GPCR complex were calculated using the Protein Contact Atlas (https://www.mrc-lmb.cam.ac.uk/rajini/index.html) using a 0.5 Å cutoff[54]. The ACKR3-VUN701 model was modified to remove all hydrogens and analyzed via the Protein Contact Atlas in an identical approach. Average contacts were manually determined. Plotting antagonist contacts along the sequence of their receptor was done using a python script.

### Reporting summary

Further information on research design is available in the Nature Portfolio Reporting Summary linked to this article.

## Data availability

The NMR solution structure of VUN701 has been deposited in the Biological Magnetic Resonance Data Bank (bmrb.io) under the accession number: 31108 and in the PDB (rcsb.org) under the accession code 8UEK. All other data supporting the findings of this study are available within the paper and its supplementary data files. PDB 7SK5 was used for this study along with nanobody PDB files listed in Supplementary Data 1. The sequences of human ACKR3 and human CXCL12α were obtained from UniProt (P25106 and P48061). The GPCR database (gpcrdb.org) was used to generate sequence conservation plot in Fig. 5a and c. Source data are provided with this paper.

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

## Acknowledgements

We thank Shawn Jenjak, Dr. Andrew Kleist, and Dr. Acacia Dishman for their comments on the manuscript figures and Julian Grosskopf for his assistance in optimizing the efficiency of the molecular dynamics simulations. This research was completed in part with computational resources and technical support provided by the Research Computing Center at the Medical College of Wisconsin. This work was supported by a National Institutes of Health Allergy and Infectious Disease grant (NIAID R37AI058072 to B.F.V.), by a National Institutes of Health General Medical Sciences grant (NIGMS R35GM133421 to J.D.M.), and by the European Union H2020-MSCA Program (Grant agreement 860229-ONCORNET2.0 for R.H. and M.J.S.).

## Author contributions

R.R.S. and B.F.V. conceived the project. B.F.V. supervised the project. R.R.S. made constructs, purified protein, and collected data. R.R.S.

performed β-arrestin assays, molecular modeling, bioinformatics, and VUN701 NMR. R.R.S. analyzed data, interpreted results, and wrote the manuscript. R.R.S. made figures. M.J.S. and R.H. co-developed VUN701 and interpreted results. F.C.P. cloned VUN701, optimized pulse sequences, and oversaw NMR data collection and analysis. JDM provided initial pcDNA constructs and input on functional assays. All authors provided comments on the manuscript.

## Competing interests

B.F.V. and F.C.P. have an ownership interest in Protein Foundry, L.L.C. and XLock Biosciences, Inc. R.H. is affiliated with QVQ Holding BV. All other authors declare no competing interests.
