## [Peer Review File · Nature Communications]

Structural Basis for Selectivity and Antagonism in Extracellular GPCR-NanobodiesREVIEWER COMMENTS

Reviewer #1 (Remarks to the Author):

See attached review

Nanobodies have become a common tool for research applications and are starting to be applied as therapeutics. This manuscript helps to establish structural guidelines for the development of nanobodies to target G protein-coupled receptors (GPCRs), one of the major classes of drug targets.

Briefly, the authors have investigated the structural requirements for inhibition of the atypical chemokine receptor ACKR3 by a nanobody (VUN701). They solve the structure of VUN701 alone, model its complex with ACKR3, and validate the model quite convincingly using mutants of both the nanobody and its target. They further provide some insights into the mechanism by which VUN701 acts as an ACKR3 antagonist and the factors contributing to its specificity for ACKR3 over related receptors. Bioinformatics comparisons put the results into a broader context, suggesting some principles that may be applied to development of nanobodies to target other GPCRs. Overall, the key conclusions are that the third complementarity-determining region (CDR3) of anti-GPCR nanobodies provide high binding affinity by interacting with the orthosteric binding site, whereas CDR1 makes a substantial contribution to binding selectivity.

In general, the data presented justify the major conclusions and these conclusions potentially will have a significant role in guiding future development of nanobody therapeutics. I have only a few points (mostly minor) that require author responses.

Main Points

1. Mechanism of ACKR3 antagonism. Based on their models, the authors hypothesize that TM7 of ACKR3 assumes two different conformational positions when bound to CXCL12 versus VUN701. They use a disulfide mutant to trap the receptor in the putative inactive (VUN701 binding) conformation and observe a reduction of activation efficacy for CXCL12, which is consistent with the model. However, I would also have anticipated a reduction in CXCL12 binding affinity (or increase in EC_{50}), which was not observed. Why not? The model would be even more convincing if they were to observe a gain of function in VUN701 binding, i.e. a decrease in the IC_{50} . Considering that CXCL12 retains some activation, this could be measured.
2. Lines 231-236. The authors propose that CDR1 is largely responsible for the selectivity of VUN701 binding to ACKR3 over CXCR3A and CXCR4. Their data with the Δ CDR1 mutant support that this region makes some contribution (~30-fold) to selectivity relative to CXCR4. However, the data for CXCR3A do not support any role of CDR1 to selectivity for ACKR3 versus CXCR3A. Understanding the role of CDR1 more thoroughly would require a more extensive mutational campaign of this region. I don't think it is required for this paper, but I would suggest toning down the statement that "receptor selectivity is largely encoded by" CDR1 interactions.
3. Since ACKR3 is an atypical chemokine receptor (not actually G protein-coupled) and its mechanism of activation is thought to be different from "typical" chemokine receptors (or other related GPCRs), it is possible that the factors enabling inhibition of this receptor also differ from these other receptors. It would be helpful if the authors acknowledge this difference and address it in the Discussion.

Minor Points

1. Line 59: "therapeutically target GPCRs"

2. Line 117: How do the data presented to this point show that VUN701 binds to an extracellular epitope?
3. Line 156: Should be Supplementary Table 6.
4. Line 161: Should be Supplementary Table 7.
5. Lines 165-6: How was solubility measured?
6. Fig 5a: Please indicate the residues corresponding to the orthosteric pocket and also label ECL1-3.
7. Fig 1a: I'm not sure I'd call this a pie chart.
8. Fig 1c, 1e and some other figures – some of the text is too small to be read at 100% zoom.
9. Lines 366-7: Please include the Cheng-Prusoff equation here.

Reviewer #2 (Remarks to the Author):

Review for “Structural Basis for Selectivity and Antagonism in Extracellular GPCR-Nanobodies” by Schlimgen et al.

GPCRs are ubiquitous, successful, and important drug targets. Despite heavy interest, many receptors cannot be suitably addressed with existing tools. One approach to address this shortcoming is the development of new tools based on antibodies. Single domain antibodies (nanobodies) present some useful properties in these efforts, although there are relatively few structurally characterized examples of nanobodies binding to GPCRs. In this manuscript, the authors present data describing the interaction of a nanobody (VUN701) with the extracellular domain of the atypical chemokine receptor (ACKR3, also known as CXCR7). Features of this interaction were used to dissect mechanistic and structural aspects of ACKR3 activation. This characterization is supplemented with analysis of previous reports of nanobody-GPCR structures to understand the features of nanobodies that are associated with GPCR binding.

Understanding how nanobodies make contact with GPCR targets is an important goal both for understanding receptor pharmacology and for designing new nanobody libraries with improved properties. The antagonist activity of VUN701 had previously been reported by some of these authors (reference 17), so the focus of this manuscript is on mechanism of action for VUN701. The characterization of the determinants of VUN701 activity (both within the nanobody and at the receptor) are detailed and robust. The authors attempt to extend these mechanistic findings to guide nanobody design more broadly, and these efforts are based on sparse bioinformatics data (only 13 examples in data set, missing some relevant examples as discussed below) that would benefit from further analysis. The sharing of models and sequence information that drive the mechanistic evaluation described in this manuscript are also needed for interested readers to effectively interpret and apply these findings.

This work is exciting and important and could be suitable for publication in Nature Communications provided the following comments can be addressed:

Major comments:

*Analysis of published nanobody-GPCR structures is missing relevant examples. It is unclear if the conclusions regarding the preferential extended structure of nanobody CDR3s will persist after analyzing these other cases. The following PDB codes correspond only to nanobodies that bind to GPCR extracellular epitopes. There may be other examples missing. The authors should take steps to ensure they are gathering all relevant examples (particularly given the sparse data set for bioinformatics analysis):

PDB: 8TH3 (<https://www.biorxiv.org/content/10.1101/2023.08.23.554128v1>); Angiotensin receptor

PDB: 6N4Y (<https://www.nature.com/articles/s41586-019-0881-4>); mGluR5

PDB: 7E6U (<https://pubmed.ncbi.nlm.nih.gov/34467854/>); CaSR

PDB: 7YM8 (<https://pubmed.ncbi.nlm.nih.gov/37339967/>); Alpha 1 adrenergic receptor

*The use of AlphaFold2 to predict antibody-antigen interactions is notoriously tricky (<https://onlinelibrary.wiley.com/doi/full/10.1002/pro.4379/>). I appreciate that the authors have provided corroborating evidence to support the utility of the model generated by AlphaFold but steps need to be taken to enable readers to independently evaluate the structures generated and evaluated. This could be done by providing the set of pdb files generated by AlphaFold2 as supporting files. Alternatively, the precise input parameters used to generate the AlphaFold2 models (including input sequences) should be provided.

*In the section on VUN701 selectivity the authors describe a single residue (F285ECL3) in ACKR3 that is not conserved in CXCR3 and CXCR4, and ascribe this difference as a prime determinant of specificity. The authors should introduce F285 into at least one of these receptors and see if it facilitates VUN701 binding.

*In Figure 4 the authors show that introducing a double A271C A293C mutation reduces beta arrestin recruitment. This reduction comes from a loss in efficacy. Given the physical setup of the assay it is unclear why using a receptor that poorly adopts a conformation that can bind arrestin would result in reduced efficacy (each receptor molecule should either recruit arrestin or not—there's no binding of half a molecule of arrestin). Could the authors confirm that this double mutant expresses at a level comparable to wild type ACKR3? If so, could they provide a hypothesis why signaling efficacy is reduced but not potency?

Minor comments:

*On line 165-166 the text says “increased VUN701 potency while also reducing solubility in PBS buffer by a factor of ~4”. The authors should describe how solubility was measured.

*The methods section description of Bioinformatic Structural Nanobody Comparison and the “all by all clustering” is lacking details. Were specific parameters used to group CDR3 structures or was this done manually? Is the length of the CDR3 a determinant of this clustering? This curated list/grouping should be published as a supporting figure or supporting file.

*A table of GPCR binding nanobody structures should be added to the SI for reference and easy lookup

*In the methods section for BRET assays the “lab of Dr. John McCorvy” is acknowledged even though he is a co-author

Reviewer #3 (Remarks to the Author):

Structural Basis for Selectivity and Antagonism in Extracellular GPCR-Nanobodies

Here are my comments.

In Line 70, Please describe in VHH small parenthesis and then use the abbreviation in rest of the manuscript.

Line 88 “Using BRET-based assays for GPCR activation, we characterized the receptor specificity of nanobody (VUN701) elicited by immunization with HEK293 cells expressing the atypical chemokine receptor 3 (ACKR3)”, meaning of the sentence is not clear and it needs to be improved.

Line 88, please write full form of BRET assay as BRET (Bioluminescence Resonance Energy Transfer) assay, and then use BRET abbreviation throughout the manuscript.

Line 125, please write full form of RMSD and NOE and then use abbreviation in whole manuscript.

Please explain significance of BRET based assay with respect of current study in two to three lines.

Citation and reference of GenScript is missing.

Line 296, please mention full form of LB and M9 medium.

Line 329, “NMR was processed” not “NMR were processed”. Please rectify the tense and grammatical mistake.

Once full form of BRET assay is mentioned in line 88, then in line 348, the heading should be “BRET Experiments”

Citation of Quikchange mutagenesis (Agilent) is missing.

Please mention full form of PBS buffer in line 165 and then start using abbreviation.

resolution: value in Å.

Line 373, what is DSSP? Write its full form and mention its significance

Line 383, please mention citation of uniprot database.

Line 398 “Structures for CKRs bound to small molecule antagonists were gathered from a literature review”. Please rectify the sentence grammatically as “Structures for CKRs bound to small molecule antagonists were gathered from literature review”.

Citation of Protein Contact Atlas is missing in whole manuscript

Line 159, please write Wild-Type instead of WT

Resolution of all the images need to be improvised

Reviewer 1

Summary

Nanobodies have become a common tool for research applications and are starting to be applied as therapeutics. This manuscript helps to establish structural guidelines for the development of nanobodies to target G protein-coupled receptors (GPCRs), one of the major classes of drug targets.

Briefly, the authors have investigated the structural requirements for inhibition of the atypical chemokine receptor ACKR3 by a nanobody (VUN701). They solve the structure of VUN701 alone, model its complex with ACKR3, and validate the model quite convincingly using mutants of both the nanobody and its target. They further provide some insights into the mechanism by which VUN701 acts as an ACKR3 antagonist and the factors contributing to its specificity for ACKR3 over related receptors. Bioinformatics comparisons put the results into a broader context, suggesting some principles that may be applied to development of nanobodies to target other GPCRs. Overall, the key conclusions are that the third complementarity-determining region (CDR3) of anti-GPCR nanobodies provide high binding affinity by interacting with the orthosteric binding site, whereas CDR1 makes a substantial contribution to binding selectivity.

In general, the data presented justify the major conclusions and these conclusions potentially will have a significant role in guiding future development of nanobody therapeutics. I have only a few points (mostly minor) that require author responses.

Main Points

1. Mechanism of ACKR3 antagonism. Based on their models, the authors hypothesize that TM7 of ACKR3 assumes two different conformational positions when bound to CXCL12 versus VUN701. They use a disulfide mutant to trap the receptor in the putative inactive (VUN701 binding) conformation and observe a reduction of activation efficacy for CXCL12, which is consistent with the model. However, I would also have anticipated a reduction in CXCL12 binding affinity (or increase in EC50), which was not observed. Why not? The model would be even more convincing if they were to observe a gain of function in VUN701 binding, i.e. a decrease in the IC50. Considering that CXCL12 retains some activation, this could be measured.

This is an interesting point. We speculate that there is no reduction in the CXCL12 potency because all crucial contacts between ACKR3 and CXCL12 are maintained. After binding to an Apo-like ACKR3, CXCL12 itself settles into the new (slightly lower) interface. Small side chain movements in the helices due to this disulfide (located outside of the orthosteric site) are not sufficient to alter binding.

The same cannot be said for the VUN701 interaction. We took the reviewer's suggestion and performed a competition assay with VUN701 and the disulfide locked ACKR3 mutant (Supplemental Figure 8). We have now shown that VUN701 had a diminished potency for the ACKR3 A271C L293C mutant, but VUN701 also resulted in a decrease in the basal β arrestin2 recruitment of this mutant. The decrease in the potency of VUN701 is due to the increased burden of VUN701 stabilizing an inactive receptor state (acting as an inverse agonist). This finding indicates that a TM7 shift is a valid mechanism of ACKR3 antagonism, but may not fully explain the full reduction in efficacy associated with VUN701 binding.

These points have been added to the text and discussed in the final paragraph of the "Mechanisms of ACKR3 Antagonism" section.

2. Lines 231-236. The authors propose that CDR1 is largely responsible for the selectivity of VUN701 binding to ACKR3 over CXCR3A and CXCR4. Their data with the Δ CDR1 mutant support that this region makes some contribution (~30-fold) to selectivity relative to CXCR4. However, the data for CXCR3A do not support any role of CDR1 to selectivity for ACKR3 versus CXCR3A. Understanding the role of CDR1 more thoroughly would require a more extensive mutational campaign of this region. I don't think it is required for this paper, but I would suggest toning down the statement that "receptor selectivity is largely encoded by" CDR1 interactions.

We have toned down the selectivity statement in the text. As evidenced by the Δ CDR1 interaction with CXCR4, CDR1 does play an important role in selectivity, however, CDR1 is not solely responsible for the selectivity of VUN701. Our modeling suggests that the N-terminus of ACKR3 plays a large role in determining selectivity, as well as the receptor's ECL2, TM5, TM6, and TM7 (as shown with new clarity in Supplemental Figure 10).

In addition to testing Δ CDR1 with CXCR3 and CXCR4, we have now tested WT VUN701 with new chimeras of CXCR3 and CXCR4 with ACKR3's ECL3 loop. If CDR1 and ECL3 alone were responsible for the selectivity, these chimeras would rescue VUN701 function. As shown in Supplemental Figure 9 (Supplemental Table 7 and 8), these chimeras are not inhibited by VUN701 - emphasizing that VUN701 selectivity is driven by several areas of the receptor in addition to ECL3. This further highlights the impressive selectivity of nanobodies at their GPCR-targets.

3. Since ACKR3 is an atypical chemokine receptor (not actually G protein-coupled) and its mechanism of activation is thought to be different from "typical" chemokine receptors (or other related GPCRs), it is possible that the factors enabling inhibition of this receptor also differ from these other receptors. It would be helpful if the authors acknowledge this difference and address it in the Discussion.

We have added this topic to our discussion. Briefly, we assert the relevance of this mechanism to typical GPCRs because VUN701 Δ CDR1 can inhibit CXCR4 (Figure 5f).

Minor Points

1. Line 59: "therapeutically target GPCRs"

Fixed.

2. Line 117: How do the data presented to this point show that VUN701 binds to an extracellular epitope?

The direct competition of VUN701 with CXCL11 and CXCL12 indicates that the ligands likely have overlapping epitopes in the orthosteric site. The linear relationship on the Schild plot verifies this information, identifying VUN701 as a reversible, competitive antagonist. These details have been clarified in the text.

3. Line 156: Should be Supplementary Table 6.

Fixed.

4. Line 161: Should be Supplementary Table 7.

Fixed.

5. Lines 165-6: How was solubility measured?

Added a description of solubility measurement to the Materials and Methods.

6. Fig 5a: Please indicate the residues corresponding to the orthosteric pocket and also label ECL1-3.

Added extracellular loop (ECL) designations to the plot. Emphasized that the small molecule ligands are all orthosteric and therefore give a nice fingerprint of orthosteric pocket interactions.

7. Fig 1a: I'm not sure I'd call this a pie chart.

Simply referred to this a "plot". Also fixed this in Figure 1b, Figure 2c, and Figure 2d.

8. Fig 1c, 1e and some other figures – some of the text is too small to be read at 100% zoom.

Enlarged the text in troublesome areas.

9. Lines 366-7: Please include the Cheng-Prusoff equation here.

Added.

Reviewer 2

Summary

GPCRs are ubiquitous, successful, and important drug targets. Despite heavy interest, many receptors cannot be suitably addressed with existing tools. One approach to address this shortcoming is the development of new tools based on antibodies. Single domain antibodies (nanobodies) present some useful properties in these efforts, although there are relatively few structurally characterized examples of nanobodies binding to GPCRs. In this manuscript, the authors present data describing the interaction of a nanobody (VUN701) with the extracellular domain of the atypical chemokine receptor (ACKR3, also known as CXCR7). Features of this interaction were used to dissect mechanistic and structural aspects of ACKR3 activation. This characterization is supplemented with analysis of previous reports of nanobody-GPCR structures to understand the features of nanobodies that are associated with GPCR binding.

Understanding how nanobodies make contact with GPCR targets is an important goal both for understanding receptor pharmacology and for designing new nanobody libraries with improved properties. The antagonist activity of VUN701 had previously been reported by some of these authors (reference 17), so the focus of this manuscript is on mechanism of action for VUN701. The characterization of the determinants of VUN701 activity (both within the nanobody and at the receptor) are detailed and robust. The authors attempt to extend these mechanistic findings to guide nanobody design more broadly, and these efforts are based on sparse bioinformatics data (only 13 examples in data set, missing some relevant examples as discussed below) that would benefit from further analysis. The sharing of models and sequence information that drive the mechanistic evaluation described in this manuscript are also needed for interested readers to effectively interpret and apply these findings.

This work is exciting and important and could be suitable for publication in Nature Communications provided the following comments can be addressed:

Main Points

1. Analysis of published nanobody-GPCR structures is missing relevant examples. It is unclear if the conclusions regarding the preferential extended structure of nanobody CDR3s will persist after analyzing these other cases. The following PDB codes correspond only to nanobodies that bind to GPCR extracellular epitopes. There may be other examples missing. The authors should take steps to ensure they are gathering all relevant examples (particularly given the sparse data set for bioinformatics analysis):
PDB: 8TH3 (<https://www.biorxiv.org/content/10.1101/2023.08.23.554128v1>); Angiotensin receptor
PDB: 6N4Y (<https://www.nature.com/articles/s41586-019-0881-4>); mGluR5
PDB: 7E6U (<https://pubmed.ncbi.nlm.nih.gov/34467854/>); CaSR
PDB: 7YM8 (<https://pubmed.ncbi.nlm.nih.gov/37339967/>); Alpha 1 adrenergic receptor
At the time of our analysis, several of these structures were not yet available. 6N4Y and 7E6U were in our original analysis but were excluded as “GPCR nanobodies” because we are specifically interested in Class A – orthosteric targeting nanobodies. We have now emphasized this point in the text.

At the reviewer’s request, we have added the newly released Nb29 (7YM8 – α 1AR), Nb2 (8FD0 - Rhodopsin), and NbE (8QOT - μ OR) to our analysis. AT118-H (8TH3 - AGTR1) is unavailable from the PDB at this time and has not been included. We note this nanobody would be a great addition (the nanobody appears to be in an extended conformation). A full list of nanobody structures with their CDR3 classification has also been added as Supplementary File 1. A full list of GPCR-targeting

nanobody structures, with information on their name, target, epitope, and CDR3 classification has been added as Supplementary File 2.

2. The use of Alphafold2 to predict antibody-antigen interactions is notoriously tricky (<https://onlinelibrary.wiley.com/doi/full/10.1002/pro.4379/>). I appreciate that the authors have provided corroborating evidence to support the utility of the model generated by Alphafold but steps need to be taken to enable readers to independently evaluate the structures generated and evaluated. This could be done by providing the set of pdb files generated by Alphafold2 as supporting files. Alternatively, the precise input parameters used to generate the Alphafold2 models (including input sequences) should be provided.

Sequence information has been added as well as more detailed instructions on how to replicate the AlphaFold modeling. PDB files used herein have also been made available in Supplementary File 3.

3. In the section on VUN701 selectivity the authors describe a single residue (F285ECL3) in ACKR3 that is not conserved in CXCR3 and CXCR4, and ascribe this difference as a prime determinant of specificity. The authors should introduce F285 into at least one of these receptors and see if it facilitates VUN701 binding. We appreciate this helpful suggestion. We have now tested VUN701 with ECL3 F285 mutants of both CXCR4 and CXCR3. Alone, this mutation is not sufficient to confer function with WT VUN701.

In addition, we have tested WT VUN701 with new chimeras of CXCR3 and CXCR4 with ACKR3's ECL3 loop (Supplemental Figure 9). These chimeras are also not inhibited by VUN701. This indicates that while ECL3 plays a significant role in enabling specificity, selectivity is also driven by several additional areas of the receptor (clarified in Supplemental Figure 10). These findings, which further highlight the impressive selectivity of nanobodies at their GPCR-targets, have been incorporated into the last two paragraphs of the "VUN701 Selectivity" section and briefly in the discussion.

4. In Figure 4 the authors show that introducing a double A271C A293C mutation reduces beta arrestin recruitment. This reduction comes from a loss in efficacy. Given the physical setup of the assay it is unclear why using a receptor that poorly adopts a conformation that can bind arrestin would result in reduced efficacy (each receptor molecule should either recruit arrestin or not—there's no binding of half a molecule of arrestin). Could the authors confirm that this double mutant expresses at a level comparable to wild type ACKR3? If so, could they provide a hypothesis why signaling efficacy is reduced but not potency?

We acknowledge and address this concern as noted in the response to reviewer 1.

These helpful comments led us to reassess our data in more detail. The A271C L293C ACKR3 mutation does impact both the expression levels and the basal β arrestin2 recruitment of this mutant. Our hypothesis as to why CXCL12 potency is unchanged while VUN701's potency decreases is now discussed in the final paragraph of the "Mechanisms of ACKR3 Antagonism". This is also pointed out by Reviewer 1 in their first major point and has been discussed there.

Minor Points

1. On line 165-166 the text says "increased VUN701 potency while also reducing solubility in PBS buffer by a factor of ~4". The authors should describe how solubility was measured.
Added a description of solubility measurement to the Materials and Methods.
2. The methods section description of Bioinformatic Structural Nanobody Comparison and the "all by all clustering" is lacking details. Were specific parameters used to group CDR3 structures or was this done

manually? Is the length of the CDR3 a determinant of this clustering? This curated list/grouping should be published as a supporting figure or supporting file.

Details have been added to the “Bioinformatic Structural Nanobody Comparison” section to enable the understanding and repetition of the bioinformatic techniques described. A full list of nanobody structures with their CDR3 classification has also been added as Supplementary File 1.

3. A table of GPCR binding nanobody structures should be added to the SI for reference and easy lookup
A table containing all nanobodies grouped by CDR3 conformation, and a table of GPCR-targeting nanobodies grouped by CDR3 conformation have been added as Supplementary File 1 and 2, respectively.
4. In the methods section for BRET assays the “lab of Dr. John McCorvy” is acknowledged even though he is a co-author
Fixed.

Reviewer 3

Summary

N/A

Main Points

1. In Line 70, Please describe in VHH small parenthesis and then use the abbreviation in rest of the manuscript.
We have added VHH in parenthesis in Line 70. While we appreciate the suggestion, we prefer to use the term nanobody rather than VHH. We do acknowledge this is an area of active debate in the nanobody/VHH field.
2. Line 88 “Using BRET-based assays for GPCR activation, we characterized the receptor specificity of nanobody (VUN701) elicited by immunization with HEK293 cells expressing the atypical chemokine receptor 3 (ACKR3)”, meaning of the sentence is not clear and it needs to be improved.
We have improved this sentence - “Using BRET-based assays for GPCR activation, we characterized the receptor specificity of a nanobody (VUN701) directed against the atypical chemokine receptor 3 (ACKR3).”
3. Line 88, please write full form of BRET assay as BRET (Bioluminescence Resonance Energy Transfer) assay, and then use BRET abbreviation throughout the manuscript.
Fixed.
4. Line 125, please write full form of RMSD and NOE and then use abbreviation in whole manuscript.
Fixed.
5. Please explain significance of BRET based assay with respect of current study in two to three lines.
Amended the “BRET Experiments” methods section to explain the significance of the BRET-based assays used in this paper.
6. Citation and reference of GenScript is missing.
Per the GenScript website – “cite GenScript as the product manufacturer.” No additional actions needed.
7. Line 296, please mention full form of LB and M9 medium.
Corrected LB to Luria-Bertani, M9 media is in its full form.
8. Line 329, “NMR was processed” not “NMR were processed”. Please rectify the tense and grammatical mistake.
Fixed.
9. Once full form of BRET assay is mentioned in line 88, then in line 348, the heading should be “BRET Experiments”
Fixed.
10. Citation of Quikchange mutagenesis (Agilent) is missing.
No citation is available for Agilent, provided product number and cited Agilent as product manufacturer.
11. Please mention full form of PBS buffer in line 165 and then start using abbreviation.
Fixed.
12. resolution: value in Å.
Unclear what this comment is referencing in the text. The VUN701 NMR structure ensemble has a backbone RMSD of 0.62 ± 0.08 Å and a heavy atom RMSD of 1.09 ± 0.08 Å (Supplementary Table 4).
13. Line 373, what is DSSP? Write its full form and mention its significance
Fixed.

14. Line 383, please mention citation of UniProt database.

Citation has been added.

15. Line 398 “Structures for CKRs bound to small molecule antagonists were gathered from a literature review”. Please rectify the sentence grammatically as “Structures for CKRs bound to small molecule antagonists were gathered from literature review”.

Fixed.

16. Citation of Protein Contact Atlas is missing in whole manuscript

Citation has been added.

17. Line 159, please write Wild-Type instead of WT

Fixed.

18. Resolution of all the images need to be improvised

Our images were compressed in the automatic conversion of the Microsoft Word document to a PDF. We have directly uploaded a PDF of this revision as well as PNGs of each figure to rectify this issue.

REVIEWERS' COMMENTS

Reviewer #1 (Remarks to the Author):

The authors have adequately addressed all my previous comments.

I think the authors have also addressed the comments of other reviewers appropriately.

Reviewer #2 (Remarks to the Author):

The authors have effectively addressed each of my comments and I recommend acceptance.

One small issue to check: there are two versions of Supporting File 3. One seems to be the ACKR3-VUN701 model structure, but I'm not sure about the other